# Genomic Characterization and Phylogenetic Analysis of SARS-CoV-2 in Libya

**Silvia Fillo** [1,*] **, Francesco Giordani** [1] **, Anella Monte** [1] **, Giovanni Faggioni** [1] **, Riccardo De Santis** [1] **, Nino D'Amore** [1] **, Stefano Palomba** [2] **, Taher Hamdani** [3] **, Kamel Taloa** [3] **, Atef Belkhir Jumaa** [3] **, Siraj Bitrou** [3] **, Ahmed Alaruusi** [3] **, Wadie Mad** [3] **, Abdulaziz Zorgani** [4] **, Omar Elahmer** [4] **, Badereddin Annajr** [4] **, Abdalla Bashein** [4] **and Florigio Lista** [1]

[1] Scientific Department, Army Medical Center, 00184 Rome, Italy; franc.giordani@gmail.com (F.G.); nellymonte88@gmail.com (A.M.); giovanni.faggioni@gmail.com (G.F.); riccardo.desantis@gmail.com (R.D.S.); nino.damore@esercito.difesa.it (N.D.); romano.lista@gmail.com (F.L.)

[2] Medical Situation Awareness Branch, General Directorate of Military Medical Service, 00184 Rome, Italy; stefano.palomba@gmail.com

[3] Public Health Emergency Office, National Center for Disease Control, Tripoli 71171, Libya; Taher.elhamdani10@gmail.com (T.H.); Ktaloa2014@gmail.com (K.T.); Atefbelkhir@gmail.com (A.B.J.); Sirajbitrou@yahoo.com (S.B.); ahmed.arusi88@gmail.com (A.A.); wadiemadi@yahoo.com (W.M.)

[4] Public Health Reference Laboratory, National Center for Disease Control, Tripoli 71171, Libya; zorgania@yahoo.com (A.Z.); oelahmer@yahoo.co.uk (O.E.); bbannajar@gmail.com (B.A.); abashein_95@yahoo.com (A.B.)

* Correspondence: silviafillo@gmail.com; Tel.: +39-06-777-039-135

**Abstract:** The COVID-19 epidemic started in Libya in March 2020 and rapidly spread. To shed some light on the severe acute respiratory syndrome coronavirus-2 (SARS-CoV-2) strains circulating in Libya, viruses isolated from 10 patients in this country were sequenced, characterized at the genomic level, and compared to genomes isolated in other parts of the world. As nine genomes out of 10 belonged to the SS1 cluster and one to SS4, three datasets were built. One included only African strains and the other two contained internationally representative SS1 and SS4 genomes. Genomic analysis showed that the Libyan strains have some peculiar features in addition to those reported in other world regions. Considering the countries in which the strains are genetically more similar to the Libyan strains, SARS-CoV-2 could have entered Libya from a North African country (possibly Egypt), sub-Saharan Africa (e.g., Ghana, Mali, Nigeria), the Middle East (e.g., Saudi Arabia), or Asia (India, Bangladesh).

**Keywords:** SARS-CoV2; SNP (single-nucleotite polymorphism) analysis; molecular epidemiology; Africa; Libya

## 1. Introduction

On 31 December 2019, an outbreak of atypical pneumonia cases was reported in Wuhan, in Hubei province of China, but the causative agent was not identified or confirmed until 9 February 2020 [1]. Once identified, the agent was named SARS-CoV-2 (severe acute respiratory syndrome coronavirus-2), and the human disease it causes was named COVID-19 [2,3]. By 24 March, when the first case of COVID-19 was diagnosed in Libya, the disease had already spread to 194 countries, with a total of 372,755 reported cases (https://www.who.int (accessed on 4 January 2021)).

The spread of COVID-19 in Libya came late. This could be attributed to several factors. First, most people live in single-family houses. Second, according to the Bureau of Statistics and Census in Libya, 68% of the population is aged 15–54 years and only 5% of Libyans are older than 65 years. The Libyan government took very early precautionary measures. In early March, the Ministry of Education adjourned all schools and universities. On 16 March, all borders, airports, and seaports were closed to avoid entry of infections from abroad. Mosques were ordered to shut down, and all social congregations were banned.

On 20 March, a curfew was imposed from 6:00 p.m. to 6:00 a.m., and only grocery stores and pharmacies were kept open during the day.

In spite of those early measures, the first case of coronavirus was reported on 24 March, a pilgrim returning from Saudi Arabia via Tunisia. The infection curve rose slowly, reaching only 75 cases by 24 May [4]. However, after that, the number of daily reported cases increased significantly, and, by 28 July, there were 3017 cases, with 579 recoveries and 67 reported deaths. Moreover, in the beginning of the infection spread, the infections were located mainly in the coastal areas and limited to the cities of Tripoli, Misurata, and Benghazi. However, in the last 2 months, it has involved more than 40 cities and towns, and, during those months, the city of Sabha in the south became the main focus of the infection nationwide.

Several published studies on the evolution of the virus, even during the short time after its appearance, have examined the genomic mutations in order to determine the path of COVID-19 diffusion in the various world regions [5–16]. In late February, the lineages characterized by specific mutations were related to particular geographical distributions and were designated as A, B, and C. Lineage A corresponds to the ancestral genome, lineage B remained predominant in China, and lineages C and A were transmitted outside China [8]. Then, on the basis of a larger number of publicly shared sequenced genomes, four distinct viral clusters exhibiting high potential to undergo global transmission were identified [16]. The four clusters were defined as super-spreader clusters 1 (SS1), 2 (SS2), 3 (SS3), and 4 (SS4). These are recognizable by the following specific signature mutations: SS1 (C8782T and T28144C), SS2 (G26144T), SS3 (G11083T), and SS4 (C241T, C3037T and A23403G). The SS clusters identified by Yang and colleagues [16] partially fit the lineages of Forster and colleagues [8]. Lineages B and C correspond to SS1 and SS2, respectively. The SS1 genomes were transmitted mainly in Asian countries (China, Vietnam, Japan, South Korea, Taiwan, and Singapore) but were also detected in North America, especially in California and Washington. SS2 was disseminated in various Asian countries, North America (United States of America (USA)), Europe, South America (Brazil), and Australia. SS3 was transmitted to several Asian countries, including Singapore and Japan, as well as Europe, USA, and Australia. SS4 was transmitted mainly to Europe, where it was responsible for the explosive increase in COVID-19 incidence in March. The first genome of this cluster was reported in Germany in late January, where the majority of SS4 strains had acquired a fourth mutation, C14408T. To explain the rapid and massive expansion of cluster SS4, mutation A23403G, which causes the D614G substitution in the S protein, was hypothesized to enhance virus fitness [17].

## 2. Materials and Methods

### 2.1. Patients, Samples, and Whole-Genome Sequencing

Fifty-seven nasopharyngeal/oropharyngeal swabs found to be PCR-positive for SARS-CoV-2 at the Libyan National Center of Disease Control were sent to the Scientific Department of the Army Medical Center in Rome, Italy (AMC) for whole-genome sequencing. Viral RNA was extracted from 125 μL of each swab sample by using the RNeasy Mini Kit (Qiagen, Hilden, Germany) and eluted in 18 μL of nuclease-free water. To confirm the positivity of the samples, RT-PCR was repeated by AMC using the Novel Coronavirus (2019-nCoV) Nucleic Acid Diagnostic Kit (Sansure Biotech Inc., Changsha, China) on an LC480 instrument (Roche Diagnostics, Mannheim, Germany). To optimize the sequencing procedures, 10 samples were selected according to the viral titer estimated from the RT-PCR cycle threshold (Ct range 16–25) and their geographical origin in Libya (Table 1).

**Table 1.** Samples analyzed in this study; [a] cycle threshold for N gene; [b] cycle threshold for ORF gene. M, male; F, female; ND, not determined.

| Internal Numbering | Sample | Gender | Age | Origin | Date | Ct N Gene [a] | Ct ORF Gene [b] | GenBank Accession Number | Coverage |
|---|---|---|---|---|---|---|---|---|---|
| 2068 | 10,230 | M | 54 | Tripoli | 13 June 2020 | 24 | 25 | MW018433 | 10× |
| 2084 | 7700 | F | 50 | Tripoli | 04 June 2020 | 18 | 20 | MW018435 | 2738× |
| 2093 | 12,156 | F | 35 | Tripoli | 19 June 2020 | 19 | 20 | MW018429 | 4017× |
| 2095 | 11,040 | F | 40 | Tripoli | 15 June 2020 | 19 | 21 | MW018431 | 964× |
| 2101 | 75 | ND | ND | Sabha | ND | 22 | 23 | MW018437 | 143× |
| 2103 | 449 | ND | ND | Sabha | ND | 19 | 20 | MW018436 | 12× |
| 2115 | 12,371 | M | 47 | Tripoli | ND | 25 | 25 | MW018428 | 43× |
| 2117 | 958 | M | 4 | Tripoli | ND | 21 | 21 | MW018434 | 557× |
| 2119 | 11,106 | M | 32 | Kabaw | 15 June 2020 | 26 | 28 | MW018430 | 11× |
| 2123 | 10,298 | M | 47 | Tripoli | 13 June 2020 | 20 | 23 | MW018432 | 1779× |

Genomic RNA was reverse-transcribed using the SuperScript III Reverse Transcriptase kit (Invitrogen, Carlsbad, CA, USA). Double-stranded DNA was subsequently synthesized by using the Klenow enzyme (Roche, Basel, Switzerland) according to the manufacturer's instructions. The Nextera XT kit was used for library preparations, and whole-genome sequencing was performed using the Illumina Miseq V3 flow cell (2 × 150 cycles) on a MiSeq sequencer following the manufacturer's instructions (Illumina, San Diego, CA, US). The reads were trimmed for quality (qscore = 20) and minimum length (100 bps) using the BBDuk trimmer integrated in Geneious Prime [18] (http://www.geneious.com (accessed on 4 January 2021)). High-quality reads were assembled by mapping to the reference genome Wuhan-hu-1 (GenBank accession number: NC_045512.2) with the bowtie2 mapping algorithm also integrated in Geneious Prime. All 10 viral genomes were deposited in Genbank (Table 1).

All the data in this study were anonymized by deleting all sensitive information. According to the Italian Data Protection Code (Legislative Decree of 30 June 2003, n. 196 ), this made it unnecessary to obtain ethics approval (Legislative Decree of 24 June 2003, n. 211—article 6).

*2.2. SNV Analysis*

SNVs (single-nucleotide variants) of each sequenced genome were obtained by applying the Find Variation/SNPs tool, a Geneious Prime feature, to the read mappings, using a minimum variant frequency of 0.85. The resulting information was integrated with amino-acid translation (Figure 1).

A set of ad hoc scripts (available on demand), written in python language and working in pipeline, was created to identify the genomes in the GISAID (global initiative on sharing all influenza data—https://www.gisaid.org (accessed on 4 January 2021)) database carrying the same mutations of the new sequenced genomes. The examined part of the GISAID database consisted of 48,782 sequences comprising all those submitted until 29 July 2020 with a collection date until 30 June 2020 that passed the filters "complete", "high coverage", and "low coverage exclusion". They are available for download from the GISAID site. For analysis, the sequences were divided in groups of no more than 3000 members, and each group was aligned with the same reference (Wuhan-hu-1, NC_045512.2) using MAFFT (multiple alignment using fast Fourier transform) v7.453 with strategy FFT-NS-1 [19]. From the resulting alignments, a script determined the mutation profile of each genome, using a uniform code related to the positions of the reference sequence. A second script compared a query SNV profile with the created SNV profile database, identifying the genomes with mutations common to the query.

**Table — Part 1**

| protein w mutation | | Nsp1_266/805 | Nsp2_806/2719 | | Nsp3_2720/8554 | | | | | | Nsp4B_8555/10054 | Nsp6_10973/11842 | | Nsp8_12092/12685 | Nsp12_13442/16236 | | | | | Nsp14_18040/19620 | | Nsp15_19621/20658 | | | |
|---|---|---|---|---|---|---|---|---|---|---|---|---|---|---|---|---|---|---|---|---|---|---|---|---|---|
| position | | 241 | 361 | 939 | 1943 | 3037 | 5359 | 6337 | 8097 | 8327 | 8420 | 8782 | 11083 | 11575 | 12223 | 13629 | 13960 | 14218 | 14408 | 15407 | 18877 | 19086 | 19955 | 20339 | 20340 | 20341 |
| ancestral allele | | C | A | A | C | C | T | G | C | C | A | C | G | C | G | C | G | G | C | C | C | G | C | T | T | A |
| mutate allele | | T | G | G | T | T | C | T | T | T | G | T | T | T | T | T | A | T | T | T | T | T | T | del | del | del |
| locus/gene | | 5' UTR | | | | | | | | | | | | | ORF1ab | | | | | | | | | | | |
| CDS position | | | 96 | 674 | 1678 | 2772 | 5094 | 6072 | 7832 | 8062 | 8155 | 8517 | 10818 | 11310 | 11958 | 13365 | 13696 | 13954 | 14144 | 15143 | 18613 | 18822 | 19691 | 20075 | 20076 | 20077 |
| Protein effect | | | None | K -> R | R -> C | None | None | None | T -> I | L -> F | M ->V | None | L -> F | None | None | None | V -> I | D -> Y | P -> L | A -> V | None | K -> N | T -> I | del | del | del |
| Codon Number | | | 32 | 225 | 560 | 924 | 1698 | 2024 | 2612 | 2688 | 2719 | 2839 | 3606 | 3770 | 3986 | 4455 | 4566 | 4652 | 4715 | 5047 | 6204 | 6274 | 6564 | | | |
| Sample | isolation loc. | | | | | | | | | | | | | | | | | | | | | | | | | |
| 10230 | TRIPOLI | T | | G | | T | | | T | | | | | | T | | | | T | | T | T | | | | |
| 958 | TRIPOLI | | | | | | | | | | | | T | | | | | | | T | | | | | | |
| 7700 | TRIPOLI | | G | | T | | C | | | T | | | T | T | | T | A | T | | | | | | | | |
| 75 | SABHA | | G | | T | | C | | | T | | | T | | | | | | | | | | | del | del | del |
| 449 | SABHA | | G | | T | | C | | | T | | | T | | | | | | | | | | | | | |
| 11106 | KABAW | | N | | T | | C | | | T | | | T | | T | | | | | | | | | | | |
| 12156 | TRIPOLI | | G | | T | | C | T | | T | G | | T | | T | | | | | | | T | | | | |
| 11040 | TRIPOLI | | G | G | T | | C | T | | T | | | T | | T | | | | | | | | | | | |
| 10298 | TRIPOLI | | G | | T | | C | T | | T | | | T | | T | | | | | | | | | | | |
| 12371 | TRIPOLI | | G | | T | | C | T | | T | | | T | | T | | | | | | | | | | | |

**Table — Part 2**

| protein w mutation | | S_21563/25384 | | | | | | ORF3_25393/26220 | | | E_26245/26472 | M_26523/27191 | ORF6_27202/27387 | | | ORF7_27394/27887 | ORF8_27894/28259 | N_28274/29533 | | | ORF10_29558/29674 | | |
|---|---|---|---|---|---|---|---|---|---|---|---|---|---|---|---|---|---|---|---|---|---|---|---|---|
| position | | 22444 | 22459 | 22468 | 23403 | 23539 | 25249 | 25563 | 25593 | 25641 | 26461 | 26735 | 27208 | 27382 | 27384 | 27863 | 28144 | 28854 | 28878 | 29449 | 29642 | 29742 | 29757 |
| ancestral allele | | C | A | G | A | A | G | G | G | G | c | C | C | G | T | T | T | C | G | G | C | G | G |
| mutate allele | | T | T | T | G | T | T | T | T | T | T | T | T | T | C | C | C | T | A | T | T | A | T |
| locus/gene | | S | | | | | | ORF3a | | | E | M | ORF 6 | | | ORF 7b | ORF 8 | N | | | ORF10 | | |
| CDS position | | 882 | 897 | 906 | 1841 | 1977 | 3687 | 171 | 201 | 249 | 217 | 213 | 7 | 181 | 183 | 108 | 251 | 581 | 605 | 1176 | 85 | | |
| Protein effect | | None | None | None | D -> G | None | M → I | Q -> H | K -> N | L -> F | L -> F | None | H -> Y | D -> Y | None | None | L -> S | S -> L | S -> N | None | TRUNCATION | | |
| Codon Number | | 294 | 299 | 302 | 614 | 659 | 1229 | 57 | 67 | 83 | 72 | 71 | 3 | 61 | 61 | 36 | 84 | 193 | 201 | 392 | | | |
| Sample | isolation loc. | | | | | | | | | | | | | | | | | | | | | | |
| 10230 | TRIPOLI | T | T | | G | T | | T | | | N | T | | T | | | T | | | | | | |
| 958 | TRIPOLI | | | T | | | | | | | | | T | | | C | C | A | | T | A | | |
| 7700 | TRIPOLI | | | T | | | T | | T | | T | | | | C | | C | A | T | T | A | A | T |
| 75 | SABHA | | | T | | | | | | | | | | | | | C | A | T | | A | | |
| 449 | SABHA | | | T | | | | | | | | | | | | | C | A | T | | A | | |
| 11106 | KABAW | | | T | | | | | | T | | | | | | | C | A | T | | A | | |
| 12156 | TRIPOLI | | | T | | | | | | | | | | | | | C | A | T | | A | | |
| 11040 | TRIPOLI | | | T | | | | | | | | | | | | | C | A | T | | A | | |
| 10298 | TRIPOLI | | | T | | | | | | | | | | | | | C | A | T | | A | | |
| 12371 | TRIPOLI | | | T | | | | | | | | | | | | | C | A | T | | A | | |

**Figure 1.** Sample single-nucleotide variants (SNVs) and AA changes. Colors: blue = super-spreader 1 (SS1) signature mutations; green = recurrent mutations in SS1 genomes; yellow = SS4 signature mutations; orange = SS3 signature mutations.

### 2.3. Phylogenetic Analysis

In silico phylogenetic analysis was performed on the newly sequenced Libyan viral genomes against three different genome datasets retrieved from GISAID until 30 June 2020. The first was related to the African high-coverage Sars-CoV-2 complete genomes, while the second and third consisted of international genomes coming from all five continents and representative of SS1 and SS4 clusters.

Raw datasets were aligned to the reference genome NC_045512.2 by using MAFFT v7.453 with default settings [19]. Then, to obtain a higher-quality dataset, the flanking regions of the aligned sequences were trimmed to the consensus range of 54 bps to 29,783 bps according to the reference sequence. Moreover, sequences containing >0.1% of N were detected with an ad hoc script (available on demand) and removed from the dataset. The final high-quality datasets consisted of 501 sequences (first dataset), 84 sequences (second dataset), and 98 sequences (third dataset). All the phylogenetic trees were calculated with PhyML v3.0 [20] (http://www.atgc-montpellier.fr/phyml/ (accessed on 4 January 2021)) on the basis of the maximum likelihood principle, with AIC (Akaike information criterion) as the substitution model [21,22], and with 200 bootstraps as the branch support test. The accession numbers of the Libyan genomes are reported in Table 1; those of all analyzed genomes are shown in dendrograms of Figures 2–4.

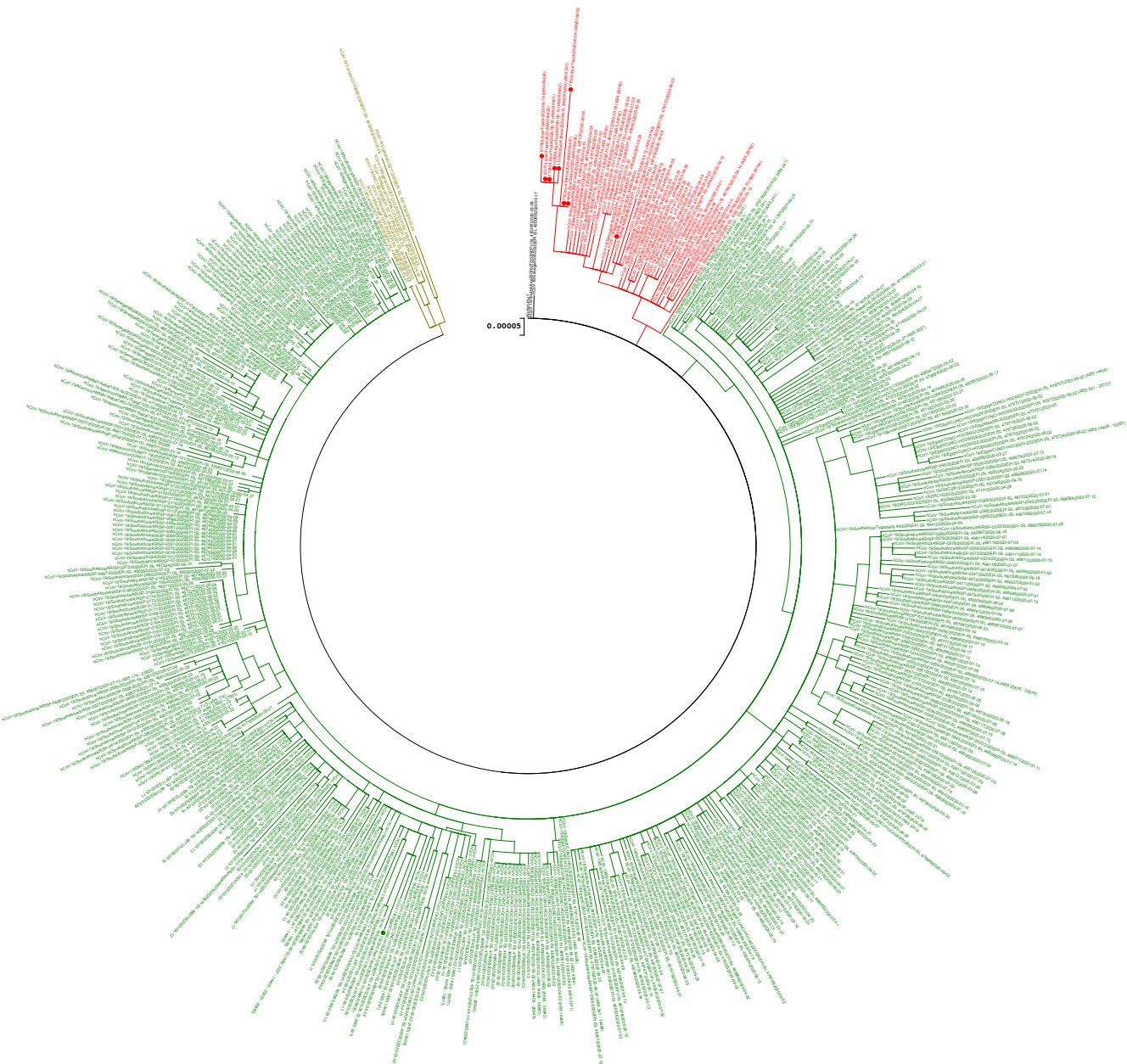

**Figure 2.** Dendrogram comparing the Libyan new sequenced genomes (marked by a dot) with a selection of representative African strains. The SNVs characterizing the main groups are shown as group names. The notation "ABS position" in brackets, adjacent to the name of a sample, means that the sample lacks the mutation in that position, albeit inserted in a group depicted as containing that mutation. Note that SS4 refers to the evolved SS4 comprising also the mutation C14408T, unless indicated otherwise. The asterisks indicate the supported clusters (bootstrap value ≥60%).

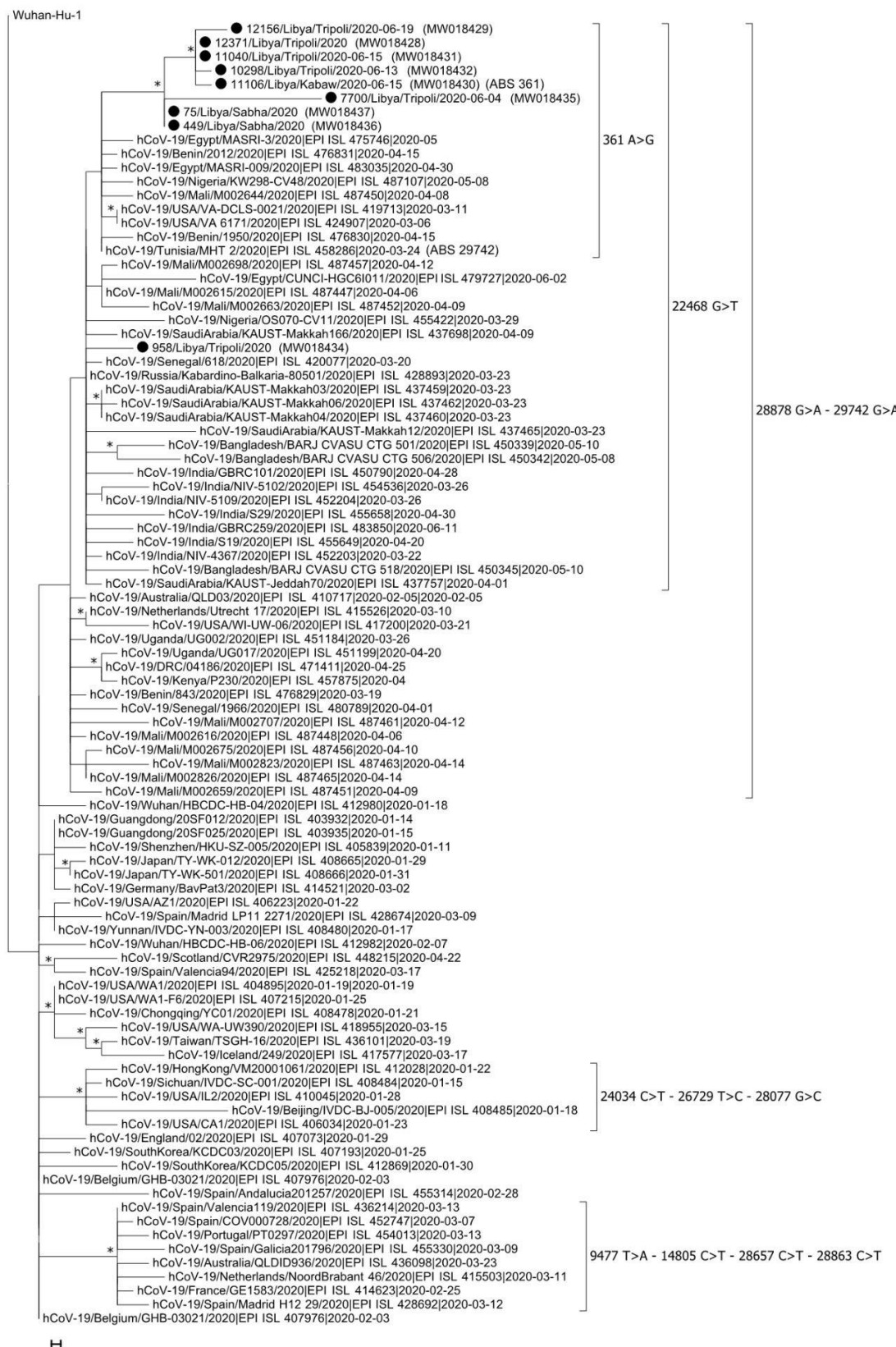

**Figure 3.** Dendrogram comparing the Libyan new sequenced genomes (marked by a dot) with international SS1 strains. The SNVs characterizing the main groups are shown as group names. The notation "ABS position" in brackets, adjacent to the name of a sample, means that the sample lacks the mutation in that position, albeit inserted in a group depicted as containing that mutation. The asterisks indicate the supported clusters (bootstrap value ≥60%).

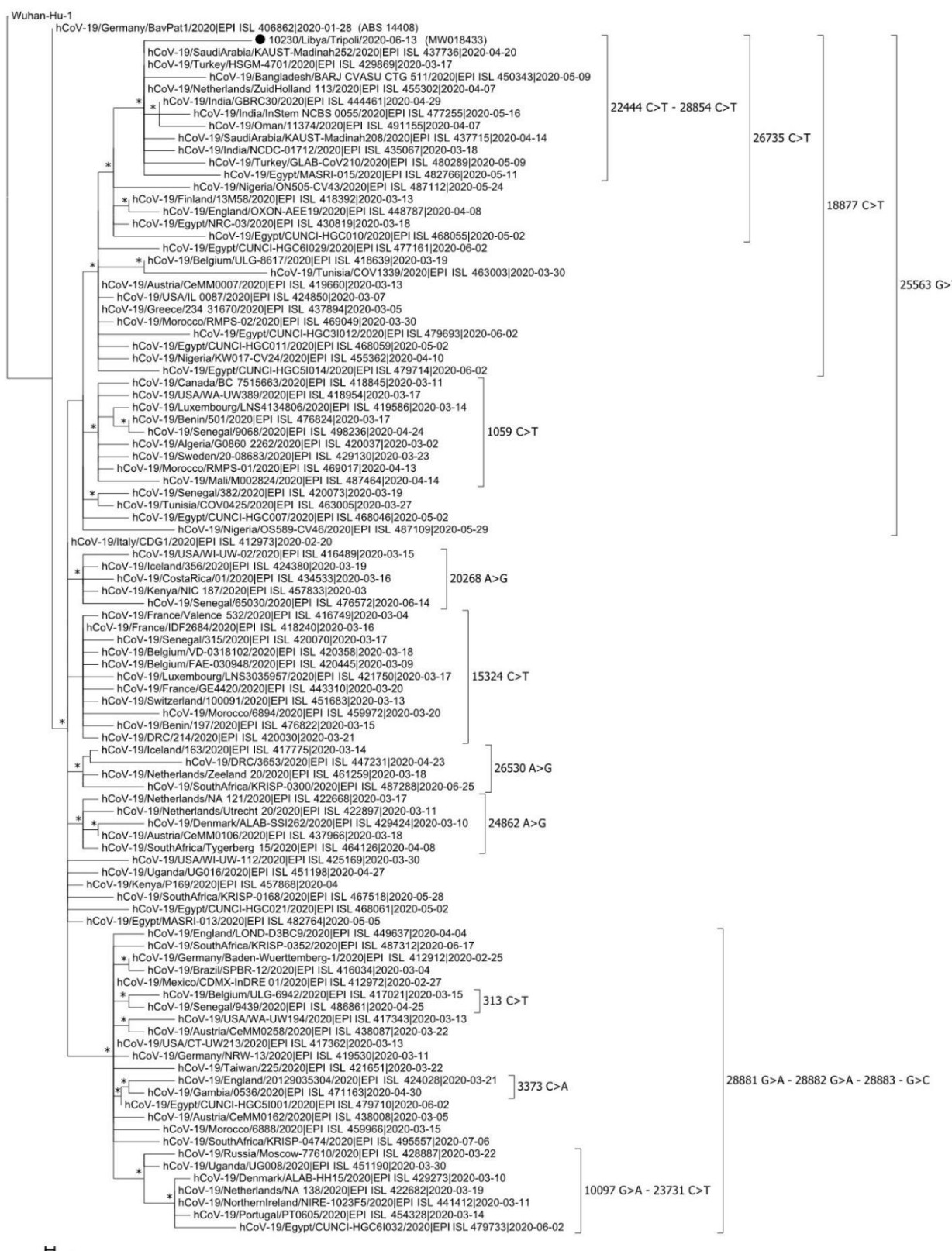

**Figure 4.** Dendrogram comparing the Libyan new sequenced genomes (marked by a dot) with international SS1 strains. The SNVs characterizing the main groups are shown as group names. The notation "ABS position" in brackets, adjacent to the name of a sample, means that the sample lacks the mutation in that position, albeit inserted in a group depicted as containing that mutation. The asterisks indicate the supported clusters (bootstrap value ≥60%).

## 3. Results and Discussion

### 3.1. SNV Analysis

To investigate the genetic diversity of SARS-CoV-2 in Libya, the 10 genomes isolated from Libyan patients living in different regions were sequenced and analyzed. Seven of these were from Tripoli, one was from Kabaw in the northwest, and two were from Sabha in the south. Genome analysis was performed relative to the Wuhan-Hu-1 reference sequence (NC_045512.2). The mutational profiles were classified following the genetic cluster scheme, suggested by Yang and colleagues, consisting of four clusters (SS1–SS4) [16]. The resulting genomic SNVs and the corresponding amino-acid positions and variations inside the proteins are shown in Figure 1. Due to their unique features, two samples (10,230 and 11,106) were included in this analysis although they had N percentage >0.1%: genome 10,230 was the only one belonging to the SS4 cluster and 11,106 was the only one isolated in Kabaw. The mutational analysis revealed 47 SNVs, of which 20 were silent mutations (synonymous), 23 were missense, three were deletions (20339delTTA20441), and one is a stop codon in ORF (open reading frame) 10. The ratio between synonymous and non-synonymous mutations was 1:1.77 across the entire set of Libyan genomes. Furthermore, 30 SNVs were found only once (63.8%), 10 were recurrent (21.3%), and seven were SS signature mutations.

During the mutational analysis, we found that nine genomes belonged to the SS1 group (signature mutations C8782T and T28144C) and one (10230, from Tripoli) belonged to SS4. This genome had the SS4 signature mutations (C241T, C3037T, A23403G) plus C14408T, as well as 11 other SNVs, of which six were missense, five produced an AA (amino acid) substitution in ORF1ab protein (K225R, T2612I, K6274N), one was in ORF3a (Q57H), one was in ORF6 (D61Y), and one was in N (S193L).

All nine SS1 Libyan strains were characterized by a haplotype consisting of the mutations G22468T, G28878A, and G29742A. Moreover, all SS1 strains except 958 also shared five other mutations; among them, A361G was the most relevant in the comparison with genomes originating in other regions, as explained below (Figure 3).

One of the SS1 genomes (7700) was different from all the others in this group. Although sharing with them 10 SNVs, it also had the SS3 mutation G11083T, a stop codon on ORF10 (K85StopCodon), and an AA mutation on E (L72F).

Moreover, the SS1 genomes also showed sporadic SNVs, some of them resulting in AA changes.

Two genomes coming from Sabha had the same SNV profile; however, in one of them (genome 75), there was a deletion of three nucleotides (20339delTTA20441). This deletion was reported in two other genomes coming from the Netherlands (EPI_ISL_455224) and the United States (EPI_ISL_485840). As these genomes did not share any other mutations with the Libyan genome 75, it seems likely that these deletions occurred independently. Mutation 20339delTTA20441 resulted in an amino-acid loss (phenylalanine) and the replacement of a serine (polar) with a cysteine (nonpolar).

We searched for mutational profiles resembling those in the Libyan genomes within a large section of the GISAID database. The search covered 48,782 sequences corresponding to those submitted until 29 July 2020, with a collection date until 30 June 2020, and passing the quality filters available in the GISAID site.

The SS1 Libyan genome haplotype (G22468T, G28878A, G29742A plus the SS1 signature mutations) was found in 80 genomes. The subgroup most similar to the SS4 Libyans (eight out of nine) was constituted by 16 genomes that also contained the mutation A361G, prevalently originating from African countries (Egypt, Benin, Mali, Ghana, Nigeria), four from Belgium (but isolated from a patient who had recently traveled to Niger: EPI_ISL_487433-36), and three from USA. Of all those, the genome with the earliest isolation date was EPI_ISL_424907 from USA (06 March 2020), followed by EPI_ISL_419713 from USA (11 March 2020) and EPI_ISL_422402 from Ghana (30 March 2020); no genome showed the basic mutational state (no mutations outside of the six characterizing the whole group), but the genomes with the lowest number of mutations (seven) were the aforementioned

three genomes with the earliest isolation date and the two genomes originating from Benin and Egypt. The remaining 66 genomes, those lacking the mutation A361G, came from India (29), Saudi Arabia (18), Bangladesh (five), Australia (one), Nigeria (four), Egypt (two), Mali (two), and Russia (two). The earliest isolated genomes were EPI_ISL_469243 from Saudi Arabia (15 March 2020) and EPI_ISL_420077 from Senegal (20 March 2020), while there were five genomes with the basic SNV: three from Saudi Arabia and two from Russia.

The most similar genome to 10,230, the only SS4 Libyan sample, originated from India (EPI_ISL_461484, isolated on 27 May 2020) and shared 10 mutations with 10,230: the four SS4 signature mutations plus C18877T, G19086T, C22444T, G25563T, C26735T and C28854T. A total of 200 sequences showed a similarity of nine mutations (those mentioned, except for G19086T), which originated mainly from Asian countries (India (144), Bangladesh (15), Oman (10), and Saudi Arabia (nine)) and more rarely from other countries (Ireland (nine), Australia (three), and Egypt (three)). Of these, the genomes with the earliest isolation dates were EPI_ISL_490003 (16 February 2020) and EPI_ISL_490004 (17 February 2020), both from Saudi Arabia, while there were eight genomes with the basic nine SNVs: five from India, two from Saudi Arabia, and one from the Netherlands.

### 3.2. Phylogenetic Analysis

To obtain a more comprehensive representation of the genetic diversity of the SARS-CoV-2 isolates included in this study, four phylogenetic trees were constructed. The first (Figure 2) compares the Libyan genomes to other African samples (501). The second and the third make comparisons with the SS1 (84; Figure 3) and SS4 (98; Figure 4) international strains, respectively. SS4 was the SS cluster gathering most African strains (445 out of 511; 87%), followed by SS1 (45; 9%) and SS2 (8; 1.6%) (Figure 2). Cluster SS1 was found in countries in the northern, southern, and central parts of the continent. All the African SS1 strains but one (EPI_ISL_418216 from Senegal) also carried mutations G28878A and G29742A. A subgroup of SS1 (26 samples) also contained G22468T in addition to the aforementioned mutations. Within this subgroup, 15 sequences also had mutation A361G, including seven Libyan sequences (however, in sample 11,106, no base could be confirmed in this position) and the other eight from Egypt, Nigeria, Mali, Benin, and Tunisia. One Libyan strain had G22468T but not A361G, while the 10 sequences with the same mutational profile originated in Nigeria (four), Senegal, Mali, and Egypt (Figure 2).

The SS4 genomes are ubiquitous in Africa. Almost all SS4 African strains analyzed here contained the C14408T SNV, which characterizes also the majority of SS4 strains in Europe [14]. Subclusters within SS4 were distinguished by the main additional mutations: the three subsequent and associated mutations G28881A, G28882A, and G28883C (156 samples), frequently present also in strains originating in other continents (America, Europe) [10], mutation G25563T (87 samples), also frequent outside Africa, especially in Europe [10], and mutations C15324T (44 samples), C16376T (38 samples), and A20268G (32 samples) (Figure 2). Libyan sample 10,230 was contained in an SS4 subcluster characterized by the previously mentioned G25563T and C18877T mutations. The other 27 members of this subcluster were prevalently from North Africa (Egypt, Morocco, Tunisia), as well as from sub-Saharan countries (Nigeria, Democratic Republic of Congo (DRC)) (Figure 2).

The SS1 international reference dendrogram (Figure 3) shows that the additional mutations found in African genomes were also disseminated in other continents. The mutations G28878A and G29742A were both present in strains isolated in Australia, the Netherlands, and USA. Strains with G22468T associated with the previous two mutations were found in Asian (India, Saudi Arabia, and Bangladesh) and European (Russia) countries.

Likewise, as the SS4 international reference dendrogram shows (Figure 4), genomes with mutations in common with Libyan sample 10,230 were disseminated outside Africa. Mutations G25563T and C18877T were present, coupled, in genomes from Europe and America (USA, Austria, Belgium, Greece, England, and Finland). Around the world, the genomes most similar to 10,230 (with shared mutations C26735T, C22444T, and C28854T, plus the previously mentioned G25563T and C18877T) originated primarily from Asian countries (India, Saudi Arabia, Oman, and Turkey), as well as from the Netherlands.

## 4. Conclusions

On the basis of the genomic features found in this study, the following can be affirmed about SARS-CoV-2 in Libya:

1. SS1 (type B) seems to be predominant in Libya (as in East Asian countries) and likely entered before the first case was reported; genome 958 had only a few SNVs in addition to the SS1 signature mutations.
2. In Libya, the SS1 lineage evolved through human-to-human transmission and acquired mutations, but the parental strain remained within the population. The SNVs increased in number from five to 10, forming a SNV pattern typical of Libyan strains.
3. Genome 7700 had the SS3 signature mutation G11083T, which was found in only one of our patients, as well as the signature mutations of SS1 and other mutations common with Libyan SS1 genomes. Thus, it can be considered an SS1 with a subsequent G11083T mutation rather than a true SS3. It probably represents the evolution of an SS1 branch already present in Libya rather than a sign of an independent entry of SS3 into Libya.
4. A second separate entrance is suggested by the finding of an SS4 genome, 10,230.

The countries from which SARS-CoV-2 entered Libya can be inferred from the regions of the world where the strains are genetically most similar to the strains isolated in Libya. Our results indicate that the SS1 Libyan genomes carrying A361G descended from a strain that was introduced from sub-Saharan or North African countries (Egypt) (Figure 3). It could also represent the evolution of a strain lacking A361G coming from a Middle Eastern country (likely Saudi Arabia) or Asia (India) (Figure 3). Libyan strain 958, which lacks A361G, could have come from other African countries or directly from Middle Eastern regions. Two strains isolated in March 2020, originating in the USA and having four mutations in common with the SS1 Libyan strain (including A361G), suggest an alternative hypothesis. Strain 10,230 of cluster SS4 was likely introduced from Asia or the Middle East. India is the most probable origin, followed by Bangladesh, Saudi Arabia, and Oman. However, given that a few strains with a high similarity to this Libyan strain have been isolated in Egypt, it is not possible to exclude an origin from adjacent African countries (Figure 4).

In conclusion, our study demonstrated the simultaneous circulation of distinct variants of SARS-CoV-2 in Libya and local SS1 lineage evolution.

**Author Contributions:** Conceptualization, S.F., F.L., T.H., K.T., A.B.J., S.B., and W.M.; methodology, S.F.; formal analysis, A.A.; investigation, S.F., F.G., A.M., G.F., R.D.S., and N.D.; resources, O.E. and A.B.; data curation, F.G. and A.A.; writing—original draft preparation, S.F. and F.G.; writing—review and editing, A.Z.; visualization, T.H. and O.E.; supervision, K.T.; project administration, A.A., B.A., and A.B.; funding acquisition, F.L. and S.P. All authors have read and agreed to the published version of the manuscript.

**Funding:** This research was funded by Italian Ministry of Defense.

**Institutional Review Board Statement:** All the data in this study were anonymized by deleting all sensitive information. According to the Italian Data Protection Code (DL 196/2003), this made it unnecessary to obtain ethics approval (art. 6 DL 211/2003).

**Informed Consent Statement:** Before participating in the present study, all patients signed the informed consent statement.

**Data Availability Statement:** All the genomes sequenced in the present study were deposited in the NIH genetic sequence database (Genbank: https://www.ncbi.nlm.nih.gov/genbank/ (accessed on 4 January 2021)) and will be publicly available once the study is published. All the other genomes analyzed in the present study are available at the genome database of GISAID (https://www.gisaid.org (accessed on 4 January 2021)), the global science initiative for genomic data of influenza and COVID-19 viruses.

**Acknowledgments:** The authors would like to thank the Public Health Laboratory reference staff, Tripoli, Libya, for their contributions and the COVID-19 Italian Army Study Group for their valuable contribution. The COVID-19 Italian Army Study Group includes Gaetano Alfano, Alessandra Amoroso, Anna Anselmo, Manfredo Bortone, Giandomenico Cerreto, Andrea Ciammaruconi, Angelo De Domenico, Nino D'Amore, De Santis Riccardo, Giovanni Faggioni, Silvia Fillo, Antonella Fortunato, Valeria Franchini, Francesco Giordani, Florigio Lista, Filippo Molinari, Anella Monte, Diego Munzi, Lucia Nicosia, Giancarlo Petralito, Maria Anna Spinelli, Stefano Palomba, Elisa Regarbuto, and Vanessa Vera Fain. We gratefully acknowledge the authors in the originating and submitting laboratories for their sequence and metadata shared through GISAID and NCBI.

**Conflicts of Interest:** The authors declare no conflict of interest.

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
