# Peer review of "Genomic Characterization and Phylogenetic Analysis of SARS-CoV-2 in Libya"

_2036-7481, doi:10.3390/microbiolres12010010_

Round 1

Reviewer 1 Report

The authors described the characterization of 10 SARS-CoV-2 strains isolated in Libya.

Snv analysis: Overall the results section needs to be cleaned up more for clarity. The mutational profile analysis is correct however I have problems reading the text. There is an unnecessary long list of mutants for each sample which makes some parts of the manuscript confusing. Authors should describe the most relevant. Furthermore, considering that the absolute frequency of mutations in the “four clusters” is very low (14, 66, 200), Table 2, 3, and 4 are redundant and should be eliminated.

Phylogenetic analysis: …“The first (Figure S1) compares the Libyan genomes to other African samples (501)”. This figure should be Figure 2, because is important for comparison with all other African samples, while Figure 2 should be deleted.  I would try plot a circular tree for 501 samples.

Minor: To be deleted:  “This likely occurred in multiple entry events despite government restrictions”….Political instability and violent clashes in this country generate constant displacement and 800 thousand people in need of humanitarian assistance.

To be deleted: “To our knowledge, no genomes belonging to the SS2 lineage have been reported in Libya at the time this manuscript was submitted”. ….is not a conclusion is you have a limited number of genomes reported in Lybia.

Reviewer 2 Report

The paper deal is very interesting and shed some light on Sars-CoV-2 spread in Lybia. However samples are not representative of the whole Country and the period taken into account: in fact 7 out 10 are from Tripoli and 6 out of 10 were sampled in June, according to the date reported in table 1. What about samples from the previuos months (March, April and May) and from other cities in Lybia? Could the Authors analyze some other samples from different areas of the Country and period (for example at the beginning of the virus entry)?

What do "Gene N" and "Gene ORF" colums represent in table 1? You should add a note 

Line 142-143: unclear

Line 157: the authors reported 48 SNVs, but in Fig 1 only 47 are reported

Line 164: replace C241C with C241T

Figure 1: why the E gene is red? what does it mean?

Round 2

Reviewer 1 Report

Figure 2: restore the original figure.